# Overcoming Treatment Resistance in Medulloblastoma: Underlying Mechanisms and Potential Strategies

**DOI:** 10.3390/cancers16122249

**Published:** 2024-06-18

**Authors:** Hasan Slika, Aanya Shahani, Riddhpreet Wahi, Jackson Miller, Mari Groves, Betty Tyler

**Affiliations:** 1Hunterian Neurosurgical Laboratory, Department of Neurosurgery, Johns Hopkins University School of Medicine, Baltimore, MD 21231, USA; hslika1@jhmi.edu (H.S.); aanyashahani@jhmi.edu (A.S.); rkwahiofficial@gmail.com (R.W.); miller.jackson2001@gmail.com (J.M.); 2Grant Government Medical College and Sir J.J Group of Hospitals, Mumbai 400008, India; 3Department of English, Rhetoric, and Humanistic Studies, Virginia Military Institute, Lexington, VA 24450, USA; 4Division of Pediatric Neurosurgery, Department of Neurosurgery, Johns Hopkins University School of Medicine, Baltimore, MD 21287, USA; mgroves2@jhmi.edu; 5Department of Neurosurgery, University of Maryland Medical Center, Baltimore, MD 21201, USA

**Keywords:** cancer stem cells, combination therapy, medulloblastoma, molecular pathways, treatment resistance, tumoral heterogeneity

## Abstract

**Simple Summary:**

Medulloblastoma is the most common type of malignant brain tumor that occurs in the pediatric population. It is associated with significant morbidity and mortality due to the long-lasting side effects associated with its treatment and its high potential for relapse despite the implementation of aggressive therapeutic modalities, including surgery, chemotherapy, and craniospinal irradiation. Hence, enhancing medulloblastomas’ response to therapy and attenuating their ability to resist treatment is an important clinical goal. Herein, we explore the various mechanisms that drive this resistance, which can be shared with other types of brain tumors, specific to medulloblastoma, or specific to a molecular subtype of medulloblastoma. Subsequently, we discuss potential targeted agents and innovative therapeutic strategies that can help in undermining the discussed mechanisms and enhancing this tumor type’s response to treatment.

**Abstract:**

Medulloblastoma is the most frequently encountered malignant brain tumor in the pediatric population. The standard of care currently consists of surgical resection, craniospinal irradiation, and multi-agent chemotherapy. However, despite this combination of multiple aggressive modalities, recurrence of the disease remains a substantial concern, and treatment resistance is a rising issue. The development of this resistance results from the interplay of a myriad of anatomical properties, cellular processes, molecular pathways, and genetic and epigenetic alterations. In fact, several efforts have been directed towards this domain and characterizing the major contributors to this resistance. Herein, this review highlights the different mechanisms that drive relapse and are implicated in the occurrence of treatment resistance and discusses them in the context of the latest molecular-based classification of medulloblastoma. These mechanisms include the impermeability of the blood-brain barrier to drugs, the overactivation of specific molecular pathways, the resistant and multipotent nature of cancer stem cells, intratumoral and intertumoral heterogeneity, and metabolic plasticity. Subsequently, we build on that to explore potential strategies and targeted agents that can abrogate these mechanisms, undermine the development of treatment resistance, and augment medulloblastoma’s response to therapeutic modalities.

## 1. Introduction

Medulloblastoma is the most frequently encountered pediatric malignant brain tumor, accounting for 6.4% of all central nervous system (CNS) tumors in children and adolescents [1,2]. An average of 318 cases of pediatric medulloblastoma (ages 0–19 years) were diagnosed yearly in the United States between 2016 and 2020 [2]. Currently, medulloblastomas are characterized as WHO grade IV embryonal tumors and are classified into four molecular subgroups. This understanding of medulloblastomas and their classification has evolved greatly over time since they were first described by Harvey Cushing and Percival Bailey in 1925. They initially characterized them as a subset of gliomas that preferentially develops from the cerebellar vermis [3,4]. Cushing and Bailey noted the presence of undifferentiated “medulloblasts” that have persisted from earlier stages of neural tube development, and hence, the nomenclature of these tumors was coined [3]. Later, further investigation into these tumors led to their identification as embryonal tumors independent from gliomas and to classifying based on their histological characteristics. Nevertheless, the 2016 World Health Organization (WHO) classification of central nervous system tumors came to revolutionize the classification of medulloblastoma into a molecular profile-based one [5]. This was reiterated in the latest 2021 WHO classification, which further expanded the molecular subclassification of these tumors. Herein, our current understanding of medulloblastomas primarily stems from a molecular perspective categorizing them as wingless-related integration site (WNT)-activated, sonic hedgehog (SHH)-activated, group 3, or group 4 medulloblastomas [6]. Nevertheless, the current mainstay of treatment of these tumors consists of maximal safe surgical resection, adjuvant multi-agent chemotherapy, and, with the exception of infants, risk-adapted craniospinal irradiation [7]. However, this multifaceted management entails several short-term and long-term adverse effects for patients and can alter their quality of life. Moreover, even with these aggressive treatment modalities, recurrence of the disease remains a major concern and the development of treatment resistance constitutes a continuously evolving threat [8]. This review aims to highlight the underlying mechanisms and processes that drive treatment resistance in medulloblastoma and augment the recurrence of these tumors. Moreover, the review explores promising strategies and tractable targets that can hinder the development of treatment resistance in medulloblastoma and may constitute a more favorable prognosis for patients. 

## 2. The Molecular Subgroups of Medulloblastoma

### 2.1. WNT-Activated Medulloblastoma

The least common molecular subgroup of medulloblastoma, WNT-activated medulloblastomas, occurs in approximately 10% of all cases [9]. These tumors stem from mutations resulting in hyperactivation of the canonical β-catenin-dependent aspect of the WNT pathway [10]. Mutations occur most commonly (85–90% of the time) in exon 3 of *CTNNB1*, which is the gene responsible for encoding β-Catenin [11,12]. This leads to degradation-resistant forms of the protein, which then accumulates in the nucleus. The other common mutation is the partial loss of chromosome 6 in an equal proportion of cases [13]. Other genetic mutations, which are not exclusive to WNT medulloblastomas, have been revealed by whole genome sequencing, most commonly, *DDX3X*, *SMARCA4*, *TP53*, and *KMT2D*. These genes are known to interact with the accumulated β-catenin and remodel chromatin, implying the development of cooperative mutations [14]. 

Tumors in the WNT subgroup typically have classic morphology, which entails round nuclei, regular cell size (less than 4× an RBC), and the absence of frequent mitosis [9]. Further, in classic medulloblastomas, Homer Wright rosettes are often present. They often originate near the midline but frequently extend into the cerebellar peduncle and brainstem, protruding through the Foramen of Luschka. These tumors have a high hemorrhage degree, possibly due to the poor development of a blood-brain barrier [15]. The enhanced permeability of the blood–brain barrier, facilitated by the release of WNT antagonists, improves the penetration of chemotherapeutic agents to the tumor site, which is important for these agents to elicit their effects. In the pediatric population, WNT medulloblastomas have favorable prognoses, however, prognosis is unclear for adult WNT medulloblastomas [13]. 

Five-year progression-free survival for this subgroup was found to be 64.4% [16]. These tumors occur most commonly in the pediatric population and are rarely metastatic. The WNT-activated medulloblastoma exhibits molecular stratification into two subgroups: WNT-α (70%) and WNT-β (30%) [17]. The former predominantly manifests in children with widespread monosomy in chromosome 6, whereas in the latter group, patients are primarily adults, and chromosome 6 remains diploid. Five-year overall survival for the WNT-α group is 97%, while for the WNT-β group it is 100% [18].

### 2.2. SHH-Activated Medulloblastoma

SHH-activated medulloblastomas are the second most common type of medulloblastomas, occurring in 30% of all patients [9]. These tumors have a bimodal distribution, wherein the two peaks of incidence are in infants and then in people older than 16 years of age [5]. In fact, it is the most abundant adult medulloblastoma type [15]. The key driver genes in medulloblastomas include *PTCH1* (28%), *TP53* (13.6–21%, primarily observed in patients aged 5 to 18 years), *KMT2D* (12.9%), *DDX3X* (11.7%), *MYCN* amplification (8.2%), *BCOR* (8%), *LDB1* (6.9%), *TCF4* (5.5%), and *GLI2* amplification (5.2%) [19,20]. *MYCN* amplification is closely linked to deletions of chromosome 9q, where the *PTCH1* gene (9q22) is also situated. While less frequent than chromosome 9 loss, SHH-activated medulloblastomas may exhibit losses in 10q and 17p, as well as gains in 3q and 9p. Mutations associated with the enduring activation of the SHH pathway in medulloblastomas are frequently detected and typically affect the SHH receptor PTCH-1 [21]. Less commonly, mutations may target suppressor of fused (SUFU) or smoothened (SMO) proteins, resulting in expression of GLI1, an oncogenic transcription factor. The specific impact of the upregulation of genes within the SHH pathway in the mechanisms of this subgroup remains unclear [22]. However, there is a suggestion that the abnormal activation of SHH signaling might trigger re-initiation of developmental programs controlling the proliferation of cerebellar external granule cells, possibly playing a crucial role in the oncogenesis of desmoplastic medulloblastomas.

Tumors in the SHH subtype typically have all histological variants, but the desmoplastic/nodular (DN) variant, at an occurrence of slightly greater than 50%, is the most common, followed by classic, and lastly large cell/anaplastic (LCA). DN is nodular, and shows neurocytic differentiation with interspliced embryonal components, wherein there is likely pericellular collagen deposition [11,23]. On the other hand, LCA is characterized by larger cell size, more cytologic variability, nuclear wrapping/molding, increased mitotic activity, and increased presence of apoptotic bodies [24]. SHH MBs primarily occur in the cerebellar hemispheres but can also occur within the cerebellar vermis. The Medulloblastoma Advanced Genomics International Consortium (MAGIC) has introduced additional substructures within SHH medulloblastomas, categorizing them into four molecular subgroups (SHH-α, β, γ, δ) [17]. These finer molecular distinctions within the SHH subtype exhibit distinct genomic irregularities and are linked to specific clinical characteristics. Notably, the SHH-δ medulloblastoma group is predominantly observed in adults, while most children are classified under the SHH-α group. Tumors found in infants are primarily composed of SHH-β and SHH-γ tumors, with the SHH-β group being associated with a slightly poorer outcome (67% 5-year survival compared to 88% 5-year survival) [25]. Currently, methylation profiling is the only way to identify the detailed subclasses of SHH medulloblastomas.

### 2.3. Non-WNT/Non-SHH: Group 3 and Group 4

#### 2.3.1. Group 3 Medulloblastoma 

Group 3 medulloblastomas are common in children and infants and comprise about 25% of all medulloblastoma cases [9]. These tumors are linked with a rapid clinical progression. They often display aggressive features, including large cell/anaplastic morphology, and may exhibit cytologic pleomorphism, augmented cell size, high mitotic activity, nuclear wrapping/molding, and the increased presence of apoptotic bodies [9]. They are characterized by specific molecular changes, such as amplification of the *MYC* oncogene and isochromosome 17q, which is known to contribute to the aggressive behavior of these tumors. In specific, Group 3γ is characterized by the presence of the *MYC* amplification and carries the worst prognosis with a 42% 5-year overall survival rate, while Group 3β is characterized by the activation of *GFI* genes, loss of *DDX31*, and gain of *OTX2* and has a 56% 5-year survival rate [18]. Finally, Group 3α commonly occurs in infants and has a high rate of metastasis at diagnosis (43%), but it is paradoxically associated with better survival (66% 5-year survival rate) [18].

Group 3 medulloblastomas are more specifically associated with midline structures within the cerebellum. They can infiltrate the surrounding structures, such as the brainstem and cerebellar peduncles. Tumors that belong to Group 3 and Group 4 medulloblastomas are typically of large cell/anaplastic nature or classic morphology, with Group 3 more likely to be of large cell histology than group 4 [9]. 

#### 2.3.2. Group 4 Medulloblastoma

Group 4 medulloblastomas are the most common in children, and the most common among all medulloblastomas, comprising 35% of cases [9]. They also occur three times as often in males as in females. Group 4 tumors tend to show rapid and progressive growth, leading to the compression of nearby structures, like the brainstem and fourth ventricle, and have the potential to metastasize to the leptomeninges, spinal cord, and supratentorial brain structures [26]. They have a poor prognosis, and often display large cell/anaplastic morphology [9]. Poor prognosis in group 4 tumors is preceded by early age at diagnosis, often infancy, as well as metastasis, while better prognosis includes loss of chromosome 11 and gain of chromosome 17. The associated predisposition genes for group 4 are also *PALB* and *BRCA2*. Just like the other groups, Group 4 is also categorized into different subgroups with distinct genetic hallmarks. Herein, the subgroups 4α, 4β, and 4γ are characterized by *MYCN* amplification, *SCNAIP* duplication, and *CDK6* amplification, respectively [27]. They all carry similar prognoses, with a 5-year survival rate of around 70–80% [18].

Group 3 and 4 medulloblastomas have the poorest prognosis of all medulloblastomas. These are both frequently metastatic non-WNT/non-SHH tumors, and exist along a transcriptomic continuum, indicating similarities between the two tumor types [26]. Where these tumors lie on this continuum is determined by molecular pathology and disease course and is thought to mirror early cerebellar development. 

Group 3 and Group 4 tumors have large intratumoral heterogeneity and the continuum can be broadly divided into eight subtypes. Subtype I is driven by *GFI1*/*GFI1B* activation and *OTX2* amplification. Subtype II is driven by *MYC* amplification, *GFI1*/*GFI1B* activation and mutations in *KBTBD4*, *SMARCA4*, *CTDNEP1*, and *KMT2D*. Subtype III is driven by both *MYC* and *MYCN* amplification, while subtype V is driven by *MYCN* amplification only. Subtype VI is also driven by *MYCN* amplification in addition to *PRDM6* activation. Subtype VII is driven by *KBTBD4* mutation and subtype VIII is driven by *PRDM6* activation and mutations in *KDM6A*, *ZMYM3*, and *KMT2C*. The drivers of subtype IV are still not known [26]. The four molecular subtypes and their characteristics are summarized in Table 1.

## 3. Main Drivers of Treatment Resistance

Medulloblastoma’s resistance to treatment is imparted by a set of anatomical, cellular, and molecular drivers that augment this tumor’s ability to evade treatment or adapt to it. Herein, we discuss the different mechanisms that drive this resistance and, ultimately, lead to an incomplete eradication of the tumor and disease relapse (summarized in Figure 1). 

### 3.1. Blood-Brain Barrier and Blood-Brain-Tumor Barrier

The blood-brain barrier (BBB) is a layer of specialized cells that surround the brain and protects it from pathogens and toxic metabolites [28]. This barrier normally serves a valuable purpose by selectively preventing unwanted foreign material from entering the brain. However, the blood-brain barrier becomes problematic during the management of different brain cancers—including medulloblastoma—because of how it can inhibit chemotherapeutic drug delivery [29]. The BBB’s impermeable nature to many different chemotherapeutic agents, therefore, presents a significant component of drug resistance [29]. 

There are many factors that make up the BBB. Firstly, there are specialized endothelial cells (ECs) that contribute to its impermeable properties with factors such as tight junctions, which are comprised of junctional adhesion molecules and a multitude of proteins [29]. Additionally, efflux transporters that are expressed on the surface of CNS ECs are another important component of the BBB. Many drugs are substrates for these efflux transporters, thereby inhibiting chemotherapeutic drug delivery to the brain [29]. There is also limited transcytosis via pinocytotic and endocytotic vesicles in the BBB and low expression of leukocyte adhesion molecules that can hinder immune cells from effectively permeating the brain [28]. In addition to these specialized EC-related factors, there are also other components such as pericytes, astrocytes, neurons, basal lamina, and extracellular matrix that are critical constituents of the BBB [28,29]. All of these cells and structures are important contributors to the BBB’s limited permeability. 

The brain’s extracellular space (ECS) also has strong connections to BBB impermeability. The ECS is made up of hydrophobic lipids, polysaccharides, and proteins that make up a complicated heterogenous environment [30]. This environment has dynamic fluid shifts influenced by metabolic activity changes and pulsating arterial blood flow [30]. Brain tumors distort standard ECS volume because of their aberrant microvasculature that is both torturous and dilated [30]. This leads to leaky capillaries and inhibited flow velocity that increases interstitial pressures in tumor tissue, which inhibits effective drug delivery [30]. There are also noteworthy enzymatic factors that contribute to BBB impermeability. Esterases, phosphatases, peptidases, nucleosidases, monoamine oxidases, and other enzymes that are coupled with cerebrovascular ECs act as metabolic barriers that are capable of deactivating or degrading drugs [30]. Both the combination of these enzymes and the ECS significantly contribute to the BBB’s impermeability. 

BBB disruption occurs with various types of brain tumors. Cancer cells can obstruct connections between the brain and ECs. This obstruction, in its turn, breaks down the BBB and creates an altered CNS structure known as the blood-brain tumor barrier (BBTB) [28]. The type of BBTB that a tumor causes is important to understand because its structure, quality, and level of function will impact treatment efficacy. For example, WNT medulloblastoma is known for its leaky BBB due to aberrant EC cell WNT signaling that causes hemorrhagic vasculature and aberrant fenestration; these factors confer a dysfunctional BBB, which is a major factor that contributes to the excellent prognosis of this molecular subtype [15,29]. On the other hand, a study of the characteristics of the BBB and BBTB in preclinical models of medulloblastoma highlighted the variability present in the permeability of these structures [28]. For instance, in genetically-engineered models of SHH medulloblastoma, SOX2+ cells were found to be major contributors to the BBTB integrity by extending processes that help in lining capillaries in their vicinity [31]. Interestingly, models of SHH medulloblastoma that were generated by xenografting rather than genetic modification displayed a more heterogenous pattern, with greater barrier integrity at the invading poles of the tumor which are in contact with normal brain tissue [28]. Similar heterogeneity was observed within patient-derived orthotopic xenografts of group 3 medulloblastoma [28]. The permeability observed in these xenograft models are attributed to the lack of a mature endothelium, interrupted coverage by astrocytes, and disrupted organization of intercellular junctions [32].

### 3.2. Genetic, Epigenetic, and Molecular Drivers of Resistance 

One prominent driver of treatment resistance in cancers is the ATP-binding cassette transporter superfamily [33,34]. This gene family has 50 members in humans that operate as membrane-encased pumps and shuttle various substrates in the body [33]. Cancer drug resistance has been linked to overexpression of drug efflux pumps in the ABC transporter superfamily, with *ABCB1* in particular being connected to chemoresistance in brain tumors such as medulloblastoma [33,34,35]. Several drugs such as etoposide and vincristine that are used to treat medulloblastoma patients are substrates of *ABCB1*, which hampers the efficacy of these chemotherapeutic agents [34]. In addition to *ABCB1*, multiple other *ABC* transporters have been linked to cancerous drug resistance such as *ABCG2*, *ABCC1*, and *ABCC2* [33,34]. 

There are many genetic factors related to drug resistance that are separate from *ABC* transporters as well. A seven-gene drug tolerant signature (*LTBP1*, *MAP1A*, *MBNL2*, *LGALS1*, *PNRC1*, *DAB2*, and *PLAAT3*) was recently discovered in multiple drug-resistant cell lines of medulloblastoma in a study regarding acquired drug resistance to cisplatin and vincristine [35]. Of all these genes, only *LGALS1* has been researched for its connection to medulloblastoma, where it is activated by the SHH pathway and upregulated in patients with the SHH medulloblastoma subtype [35]. Another gene set comprising of *ATR*, *LYK5, MPP2*, *PIK3CG*, *PIK4CA*, and *WNK4* has been connected to increased proliferation and cisplatin resistance in medulloblastoma, particularly with overexpressed *LYK5* and *PIK3CG* [36]. Specifically, it has been noted that the inhibition of kinase p110γ (which is encoded by *PIK3CG*) increases medulloblastoma sensitivity to cisplatin, and that the p110γ phosphoinositide 3-kinase isoform can serve as an effective target for medulloblastoma therapy [36]. There are also different types of genes such as members of the metallothionein (MT) gene family that have been connected to drug resistance in medulloblastoma for chemotherapeutic agents like carmustine (BCNU) [37]. Upregulated members of the MT gene family (*MT2A*, *MT1X*, *MT1L*, *MT1A*, *MT1E*, *MT1F*, *MT1H*, *MT1B*) code for low weight and thiol-rich proteins that have been associated with chemoresistance; mechanistically, thiol groups in MTs are known to form covalent bonds with the electrophilic centers of various therapeutic drugs like chlorambucil, melphalan, cyclophosphamide, and mechloroethamine [37]. BCNU and its relative decomposition products are known to have similar types of electrophilic centers to the previously listed drugs, likely indicating that the observed BCNU resistance may stem from this same type of drug-deactivating covalent linkage [37]. 

There are also several other known genetic drivers of resistance. Gene mutations of *SUFU* (a SHH-pathway regulator) are a noteworthy driver of drug resistance in medulloblastoma, as they have been observed to induce resistance to the SMO inhibitor, vismodegib [38]. Additionally, abnormal *TP53* gene expression is another genetic driver involved with over half of all human cancers, and *TP53* mutations can cause significant changes that alter genetic and protein-related activities [39,40]. Increased *TP53* expression is connected to chemoresistance in medulloblastoma, and it has been noted that p53 suppression boosts drug sensitivity in multiple types of medulloblastoma [40]. The inhibition of melanoma antigen (*MAGE*) and G antigen (*GAGE*) genes have also been connected to increased apoptosis of medulloblastoma cells and sensitivity to certain chemotherapeutic drugs like cisplatin and etoposide, suggesting that they play a notable role in chemoresistance [41]. *MAGE* and *GAGE* genes belong to the cancer/testis associated protein family (CTA), which is associated with cell proliferation, differentiation, and survival [42]. Additionally, microRNA dysregulation is another gene-related factor that was identified as a primary driver of medulloblastoma chemoresistance in a recent study [43]. Abnormal SHH signaling activation can arise from SMO inhibition via dysregulated miRNAs binding to the 3′ untranslated regions of SMO miRNA [43]. Specific types of miRNA upregulation in granule progenitor cell differentiation can also inhibit growth and allow for cell maturation, as well as target the downstream oncogene *GLI1* [43]. These factors suggest that dysregulated miRNAs can abnormally activate SHH signaling and confer chemoresistance. 

Epigenetic factors can play important roles in drug resistance as well. Multiple oral DNA alkylating drugs like temozolomide and lomustine have activity levels that are suppressed by the overexpression of *O6*-methylguanine-DNA-methyltransferase (MGMT), which is a feature of many medulloblastoma tumors in patients as well as a resistance mechanism that inhibits the therapeutic efficacy of both drugs [34]. Furthermore, H3 lysine 27 trimethylation (H3K27me3) is another epigenetic mechanism associated with multi-drug resistance in various types of cancers, and it has also been connected to medulloblastoma radiotherapy resistance [44,45]. Separate methylation mechanisms such as aberrant CRABP-II methylation have also been identified as resistance-causing factors; in the case of CRABP-II, aberrant methylation was found to induce resistance to the anti-cancer agent retinoic acid [46]. 

Molecularly, there are numerous factors that contribute to drug resistance in medulloblastoma. Y-box binding protein 1 (YB-1) is a protein that has been associated with chemoresistance [35]. YB-1 has been connected to most mRNA and DNA-related cell processes including DNA repair, proliferation, and replication [35]. YB-1 gene expression is associated with medulloblastoma mortality and seems to indirectly regulate cell death-related gene transcription and inflammatory activity, as well as being involved in cell invasion [35]. YB-1 may also serve as an interacting factor for MYC oncoprotein, which further details the important role that YB-1 plays in the progression of medulloblastoma [35]. There is also evidence that *YB-1* knockdown increases drug sensitivity for numerous chemotherapeutic agents and that it is linked to cellular resistance mechanisms in medulloblastoma, such as *ABCB1*-related resistance [35]. Additionally, there are other proteins, such as the anti-apoptotic BCL-2 family member proteins, that mediate resistance mechanisms in medulloblastoma [34]. Members of the BCL-2 family such as BCL-XL and MCL-1 have been specifically highlighted for their chemoresistance properties in medulloblastoma, since anti-apoptosis is a key characteristic of cancer that is related to chemoresistance [34,47]. Furthermore, mutations of certain proteins are known to confer drug resistance in medulloblastoma as well; for example, SMO mutations resulting from use of the Hedgehog inhibitor GDC-0449 can disrupt the binding of drugs and induce drug resistance [48]. 

There are other molecular factors that contribute to treatment resistance. For instance, inhibition of Estrogen Receptor β (ERβ), a supplemental therapeutic strategy, decreases cisplatin efficacy against medulloblastoma cells [49]. ERβ has also been connected to medulloblastoma cell growth and motility in addition to its chemoresistant effects with cisplatin treatment [49]. Calcium-tissue concentrations are another molecular factor that have been connected to medulloblastoma therapy resistance, with abnormal cytosolic Ca^2+^ activity being characterized as a driver of resistance [50]. Additionally, p90 ribosomal S6 kinase (RSK) inhibition has been found to circumvent SMO resistance, suggesting that RSK plays an important role in the development of this resistance [51].

### 3.3. Overexpression of Alternative Pathways/Downstream Effectors

There are numerous pathways besides the WNT and SHH pathways that are upregulated in medulloblastoma. Firstly, the IL-6/STAT3 pathway has been associated with tumorigenesis and acquired resistance in Group 3 MBs specifically [52]. STAT proteins are associated with multiple types of cancer, with STAT3 phosphorylation occurring downstream of Janus Activated Kinases (JAKs) being associated with cancerous drug resistance [52]. IL-6 is a critical cytokine for tumorigenesis that serves as an upstream regulator of STAT3 and is closely involved with STAT3 phosphorylation and activation [52]. 

Moreover, the NOTCH network is also an important player in medulloblastoma drug resistance. Normally, the NOTCH network is involved with embryonic development and tissue homeostasis [53]. However, it also has direct connections to chemoresistance because of its immune response and microenvironment maintenance roles for tumors, thereby making it a critically important pathway related to chemoresistance [54]. The NOTCH signaling pathway helps balance the ratio of cell proliferation to apoptosis [55]. A skewed balance in favor of proliferation over apoptosis that occurs with NOTCH dysregulation is undoubtedly connected to tumorigenesis and drug resistance, as inhibited apoptosis is associated with therapy resistance [47]. 

The mTOR pathway is another key pathway and known mechanism of resistance for medulloblastoma [56]. mTOR is a critical coordinator of cellular growth and cross-talks with the IDO1 pathway, leading to immunosuppression and induced resistance [56]. The mTOR pathway has numerous activators such as kinases p110γ and LYK5 that play into proliferation and chemoresistance [36]. It is also noteworthy that the IGF/PI3K pathway, which normally serves a critical role during neonatal and pubertal development, have been connected to proliferation and resistance in multiple types of cancer including medulloblastoma [57]. Furthermore, the activation of several other pathways such as RAS/MAPK and the AKT/PIK3 pathway have been found to cause SMO inhibition resistance in an SHH cell line [58]. Specifically, PI3K signaling boosts SMO expression and its corresponding downstream effector GLI2, and it has been shown that PI3K inhibition has slowed the development of treatment resistance with medulloblastoma [58]. Additionally, the previously mentioned protein SUFU is a downstream effector of SMO in the SHH pathway along with *GLI1*. Herein, activating mutations of either *SUFU* or *GLI* can confer notable resistance to SMO inhibitors [54].

### 3.4. Cancer Stem Cells

The role of cancer stem cells in the progression, aggressiveness, metastatic behavior, and resistance to treatment of several tumor types has been heavily studied and elucidated in the literature [59,60,61,62]. Cancer stem cells are characterized as a subset of tumor cells that have an inexhaustible self-renewing capacity that can reestablish the tumor and drive recurrence [63]. This subset of cells also maintains a multipotent undifferentiated state that has the potential to give rise to various differentiated progeny [63]. Additionally, cancer stem cells are privileged by a set of characteristics that makes them more resistant to therapeutic modalities. These characteristics include quiescence, expression of drug efflux transporters, augmented ability to scavenge reactive oxygen species (ROS) and repair DNA damage, overactivation of protective developmental and stemness pathways, and enhanced ability to cope with stressful microenvironmental conditions (hypoxia, nutrient deprivation, inflammation, etc.) [64].

In this context, medulloblastoma stem cells (MBSCs) have attracted significant attention as primary contributors to treatment resistance and recurrence. One of the important avenues in this domain was the characterization of the specific signatures that are unique to MBSCs and that set them apart from the remaining medulloblastoma cells within the tumor. One of the most popular stem cell markers is CD133, also known as prominin-1 (PROM1). The importance of this marker as an identifier of MBSCs was first elucidated in 2003 [65], when Singh et al. demonstrated the self-renewal capacity of CD133+ medulloblastoma cells which was limited in CD133− cells. Further studies emphasized the role of CD133, the high expression of which also correlates with worse prognosis for patients with medulloblastoma [66]. In addition, colonies expressing CD133, as well as other stemness markers such as Nestin, were found to exhibit resistance to radiation therapy and treatment with TNF-related apoptosis-inducing ligand (TRAIL) [67,68]. 

Alternatively, in the PTCH haploinsufficient (PTCH+/−) model of SHH medulloblastoma, CD133+ cells do not appear to be a major population of self-renewing stem cells. Instead, CD15+ cells were the colony that exhibited a multipotent self-regenerating potential [69]. Another marker that is specifically important in PTCH+/− SHH medulloblastoma is SOX2. In fact, SOX2+ MBSCs offer a tie-in with the neurodevelopmental origins of medulloblastoma tumorigenesis, because these cells constitute a transient subpopulation that gives rise to the external granular layer during cerebellar development [70]. However, the persistence of SOX2+ cells in SHH-activated medulloblastoma is associated with worse prognosis, resistance to the SMO inhibitor vismodegib, and resistance to antimitotic chemotherapeutic agents [71]. Finally, a third signature that has been shown to play critical roles during the different phases of SHH-activated medulloblastoma development is OLIG2. Specifically, OLIG2+ cells are enriched in the initial phase of tumor establishment and growth, then they regress into a quiescent population in the mature tumor [72]. Nevertheless, these cells are reawakened during recurrence and have the ability to reestablish the tumor and drive treatment resistance [72]. 

On the other hand, despite their critical role in driving SHH-activated medulloblastoma initiation and recurrence, neither SOX2 nor OLIG2 were found to be correlated with poor outcomes in Group 3 medulloblastoma, highlighting a role for a different population of stem cells. Indeed, SOX9 expression correlates with poor prognosis in Group 3 medulloblastoma, and SOX9+ MBSCs have been shown to be enriched in relapsed and treatment resistant tumor subpopulations [73]. 

When it comes to specific molecular pathways that contribute to the resilience and treatment resistance of MBSCs, several effectors have been investigated and proven to contribute in different ways. For instance, the Notch pathway is implicated in the response of MBSCs to hypoxic conditions and the hypoxia inducible factor (HIF)-1α-induced maintenance of the undifferentiated status [74]. In addition, radiation therapy has been proven to result in the overactivation of the PI3K/Akt pathway in MBSCs as an adaptive mechanism, while inhibiting this pathway resulted in a greater sensitization of these stem cells to radiation therapy [75].

### 3.5. Intratumoral and Intertumoral Heterogeneity

The clonal heterogeneity of tumors has long been studied as one of the major contributors to treatment resistance in cancer and one of the main reasons behind the failure of certain targeted therapies despite preclinical promise. The revolution in ability to profile the molecular, genetic, and epigenetic landscape at the single cell level brought the importance of intra and intertumoral heterogeneity into the light, and since then, great leaps have been made to discover this heterogeneity at various levels [76]. In this context, Hovestadt et al. characterized the different metaprograms that dominate the transcriptional landscape of each subtype of medulloblastoma (Figure 2). Herein, they identified four cellular states within WNT-activated MB, and each of these states was related to cell-cycle progression, protein biosynthesis and metabolism, neuronal differentiation, or the WNT pathway itself [77]. Cells within WNT-activated MBs would perform variably on each of these axes. In fact, the same study found that a niche of cells that had high protein biosynthesis and metabolism capacity but performed low on the neuronal differentiation and WNT pathway axes was a major driver of tumor growth and survival. As for SHH-activated MB, the transcriptional metaprograms present with SHH-activated MBs were cycle-associated, early neurodevelopmental progenitor-like, and more differentiated neuronal-like [77]. Similarly, three metaprograms were identified for group 3 and group 4 MBs, which are related to cell cycle progression, undifferentiated progenitors, and differentiated neuronal-like states. Interestingly, group 3 tumors had a predominance of malignant cells that manifested the undifferentiated state, whereas group 4 tumors had a predominance of those that exhibited a more differentiated profile [77]. 

Riemondy et al. also utilized single-cell RNA sequencing to investigate the intratumoral heterogeneity within each of the different subtypes. They were able to identify a greater number of cellular states within each subtype; however, they still belonged to the same metaprograms characterized previously [78]. Riemondy et al. not only corroborated the tumoral heterogeneity at the malignant cell level, but also explored this heterogeneity at the level of the tumoral microenvironment (TME). In specific, they identified separate cellular states and programs that characterize the immune cells present in the TME [78]. Beside the classical classification into lymphoid and myeloid cells, they further classified myeloid cells into two populations. One was named the complement myeloid (complement-M) population, which shows a substantial expression of the subunits related to complement component 1q [78]. This complement system is implicated in the elimination of synapses during neural development [79]. Another was identified as a population of myeloid cells that overexpresses phagocytosis markers that are closely associated with the immunoregulatory M2 polarization. This population was named the M2-activated myeloid population (M2-M) [78]. These two populations show a differential distribution within different subtypes of MB, with the SHH-activated subtype being notorious for the abundance of M2-M population [78]. The TME heterogeneity is not limited to that of myeloid cells but is also observed at the level of T-cells and non-immune components of the environment, such as fibroblasts [80]. As expected, these differences in the TME have implications when it comes to responses to therapy. For instance, Pham et al. found that, in a murine medulloblastoma model, group 3 medulloblastoma tumors have a greater infiltration of PD-1+ CD8+ T-cells as compared to SHH-activated medulloblastoma [81]. This observation translated into an augmented effect for the immune checkpoint inhibitor, anti-PD-1, in group 3 tumors [81].

Another important type of heterogeneity that significantly impacts treatment response is the intertumoral heterogeneity that exists between primary and recurrent medulloblastomas. Clinically, recurrent tumors or metastases are usually treated based on the assumption that they carry the same characteristics as their primary tumors. The molecular and genetic profiles of recurrent tumors are hard to confirm, since they are not frequently resected or biopsied at the time of recurrence due to associated morbidity and complications [82]. In this context, Ramaswamy et al. confirmed that the molecular subtype of medulloblastoma at recurrence is consistent with that of the original tumor [83]. This might be related to the fact that different molecular subtypes have different neurodevelopmental origins [27]. However, despite this lack of subtype switching, the recurrent tumors display significant divergence from the distribution of cellular states and genetic footprint of the primary tumor [82]. This change in the tumor’s clonal architecture and the rise of resistant colonies during recurrence hinders the most effective response to treatment. Hence, a suggested method to prevent this is preemptively targeting the molecular pathways that are responsible for driving resistance in the initial management of the primary tumor [82]. However, further studies are needed to select the most effective targets and validate the efficacy of this strategy.

### 3.6. Metabolic Plasticity

The concept of metabolic reprogramming in cancer cells has been around since Otto Warburg highlighted the extensive reliance of cancer cells on glycolysis without oxidative phosphorylation even in the presence of oxygen, which is known as “aerobic glycolysis” or “the Warburg effect” [84,85] and is considered one of the hallmarks of cancer [86]. However, the impressive plasticity of this reprogramming and its various contributions to cancer initiation, progression, and resistance has attracted significant attention over the past decades. Indeed, metabolic rewiring can impact cancer cells’ response to hypoxia, oxidative stress, and other anti-tumor effects achieved by therapeutic agents and subsequently contribute to the tumor’s resistance to these agents [87,88]. In specific, the landscape of metabolic reprogramming has been explored in the context of medulloblastoma even beyond the scope of glycolysis, and now includes evidence on plasticity in a myriad of metabolic pathways, such as glutaminolysis, lipid metabolism, amino acid metabolism, and nucleotide synthesis pathways [89]. In fact, the different molecular subtypes of medulloblastoma were found to have different metabolic signatures [90]. For instance, major prognostic pathways in group 3 medulloblastoma were those related to tyrosine metabolism and the pentose phosphate pathway, while the serine synthesis pathway and one-carbon cycle were significant players when it comes to the prognosis of SHH-activated and group 4 MBs [90]. 

Recently, more attention has been diverted towards studying the metabolic changes that can occur in medulloblastoma in response to treatment and how this dynamic rewiring contributes to resistance. For instance, Sun et al. generated a model of radioresistant medulloblastoma cells from the ONS-76 cell line. The resistant cells exhibited low mitochondrial respiration and subsequently maintained a low ROS level. They also had higher lactate production and existed in a low energy state as compared to their parental cells [91]. Treating the cells with dichloroacetate (DCA), which can disrupt the metabolic patterns established by the resistant colonies, resulted in higher ROS production, increased sensitivity to radiation therapy, reversal of the stemness characteristics of the cells, and reduction in their DNA repair ability [91]. This portrays the importance of the metabolic adaptations in generating and maintaining resistance to therapy. Additionally, Bakhshinyan et al. studied the differential gene expression in a patient-derived xenograft model of recurrent therapy-resistant medulloblastoma and highlighted several pathways and cellular processes that drive therapeutic resistance and relapse. Among these pathways, oxidative phosphorylation and lipogenesis appeared to be overexpressed in the refractory model of medulloblastoma [50]. Interestingly, lipid homeostasis has been implicated in the resistance of SHH-activated medulloblastoma to CDK4/6 inhibitors. Herein, the endoplasmic reticulum stress that is induced by CDK4/6 suppression promotes the production of SMO-activating sterol lipids [92]. These lipids in their turn drive sustained SHH signaling, cancer cell survival, and subsequent resistance to treatment [92]. These studies, although few in the context of medulloblastoma, open the door for further investigation into the contribution of metabolic rewiring to treatment resistance and how targeting this process can be leveraged to circumvent recurrence and augment the action of existing therapeutic modalities.

## 4. Therapeutic Modalities to Overcome Resistance

### 4.1. Disrupting the Blood Brain Barrier 

As elaborated previously, WNT-activated medulloblastoma is the only subtype with a ‘leaky’ BBB and BTB [31]. The remaining medulloblastoma subgroups, much like other CNS malignancies, present a major challenge to therapy resulting in poor drug penetrance through these barriers [15]. Chen et al. hypothesized that SOX2 positive cells are the main constituents of an impermeable blood tumor barrier in medulloblastoma. SOX2 is also a major stemness factor in SHH-activated medulloblastoma. Further investigations identified Piezo 2 as a modifier of SOX2 expression. Knockdown of Piezo 2 in mice medulloblastoma models resulted in decreased expression of the quiescent phenotype of SOX2+ cells and increased permeability of etoposide through the BTB. Additionally, it was observed that SOX2+ cells sheathe the capillary endothelial cells and can directly influence its morphology via the WNT/*β* catenin pathway [93,94]. 

A popular method that is used for increasing drug permeability to the brain is encapsulating them in nanoparticles (Figure 3). In this context, the use of fucoidan-based nanoparticles that are targeted towards P-selectin for the delivery of vismodegib has shown impressive efficacy in delivering the drug in a selective manner and improving its permeability across the BBB [95]. Moreover, non-targeted nanocarriers can also achieve augmented crossing of drugs through the BBB. Herein, Hwang et al. loaded vismodegib into polyoxazoline block copolymer micelles, which led to an enhanced delivery of the drug and significantly superior survival in a medulloblastoma mouse model as compared to untreated controls and mice that received free vismodegib [96]. The same group also found that the encapsulation of the CDK4/6 inhibitor, palbociclib, into polyoxazoline nanoparticles resulted in an improvement in the delivery of the drug and a significant survival advantage in a mouse model of an aggressive SHH medulloblastoma [97]. 

Recently, attention has been directed towards harvesting the potential of focused ultrasound (FUS) technology. In particular, low intensity FUS has a wide set of applications, primarily BBB disruption, improved drug delivery, and immune response enhancement [98]. Microbubbles are an important component for energy transfer from these low intensity ultrasound waves to target tissues [99] (Figure 3). For instance, the delivery of siRNA molecules engineered for *SMO* knockdown in preclinical group II medulloblastoma models using microbubble-enhanced FUS demonstrated increased blood-brain barrier permeability and effective drug delivery to target [100].

Additionally, the use of image-guided drug delivery has also emerged as an appealing avenue for enhanced theranostic targeting and to complement the effects of nanomedicine. Several applications have been suggested for image-guided drug delivery, including utilizing it to enhance the delivery of nanoparticles or to predict the response to nanotherapeutic approaches [101,102]. For instance, combining the drug-loaded magnetic nanoparticles, which can be guided to the brain using MRI, with targeted focused ultrasound to disrupt the BBB has shown enhanced drug delivery and demonstrated the ability to track drug distribution in the brain [103].

### 4.2. Targeting Genetic, Epigenetic, and Molecular Drivers of Resistance

Several efforts have been channeled towards targeting the different drivers that promote the development and maintenance of treatment resistance in medulloblastoma. In this context, Nakata et al. showed that epigenetic targeting via the histone deacetylase (HDAC) inhibitor RG2833 in WNT and SHH medulloblastoma results in hypomethylation of the *Schlafen11* gene and overproduction of SLFN11, which makes tumor cells vulnerable to cisplatin treatment as confirmed in rodent models [104]. Similarly, epigenetic regulation of MYC through pharmacological and genetic inactivation of the upstream Ableson tyrosine kinase (ABL1/2) results in decreased C-MYC expression and tumor progression. ABL inhibition also downregulates the epithelial mesenchymal transition (EMT) to restrict leptomeningeal dissemination of tumor cells [105]. Protein arginine methyltransferase 5 (PRMT5) has also been recognized as an epigenetic promoter and stabilizer of MYC in various cancers [106]. Clinical trials are ongoing to assess the efficacy of PRMT inhibitors in breast cancer (NCT04676516), gliomas (NCT04089449) and various hematological malignancies (NCT03886831, NCT03573310). TNG908 has been reported as a potent PRMT5 inhibitor with the ability to cross the blood-brain barrier in recent trials (NCT05275478). SSRP1, a subunit of the histone-chaperone FACT complex, has emerged as another promising drug target in MYC amplified medulloblastoma. Curaxin drug CBL0137, a FACT inhibitor, displayed acceptable safety levels and anti-tumor effects in a recent phase I clinical trial (NCT01905228) [107,108]. On the other hand, radio-resistance develops in group 3 and 4 medulloblastoma by means of activation of pathways promoting DNA repair and cell proliferation. EZH2 (enhancer of zeste 2 polycomb repressive complex 2 subunit) dependent degradation of trimethylated H3K27 is a major pathway in this mechanism. The H3K27me3-deficient medulloblastoma cells express overactivity of the PI3K/AKT pro-survival pathway. Upstream inhibition of AKT signaling or BET mediated silencing of EZH2 via JQ1 are potential therapeutic targets in radiation resistant, relapsing medulloblastomas [44].

The mTOR pathway is also frequently associated with all subgroups of medulloblastoma. In medulloblastoma cell lines, a rapamycin-dependent increase in indoleamine 2,3-dioxygenase 1 (IDO1) contributes to the immune surveillance escape of tumor cells. IDO1 catalyzes amino acid degradation, and this change in the microenvironment interferes with mTOR autophagy signaling. IDO1-induced FOXP3 upregulation is not completely understood, but the increased T_reg_ population in the tumor neurosphere contributes immune tolerance to these cells [56]. Thus, IDO1 antagonists can be considered as novel drug candidates. Moreover, mTOR inhibitor resistance is an emerging obstacle in cancer treatment and can be overcome by administering multi-drug combination therapy with second generation mTOR inhibitors and PI3K/AKT inhibitors [58].

Vismodegib, an SMO inhibitor, is the most widely used drug in SHH-activated medulloblastoma. However, resistance usually develops due to SMO mutations and epigenetic or downstream molecular target variations in the hedgehog pathway. Moreover, first generation SMO antagonists are usually not active in medulloblastoma subtypes harboring SUFU mutations and MYCN/GLI2 amplifications. Sonidegib, a second-generation SMO inhibitor, is currently under phase I/II clinical trials in recurrent and highly metastatic SHH amplified medulloblastoma (NCT01125800, NCT01708174). Additionally, HhAntag (a benzimidazole) and the bis-amide class of SMO inhibitors was able to exert inhibitory effects even against vismodegib-resistant SMO both in vitro and in vivo [109]. Clinical trials targeting inhibition of downstream targets like GLI in SMO-mutated resistant medulloblastoma via Silmitasertib, a potent and selective casein kinase 2 inhibitor (NCT03904862) and Samotolisib, a PI3K inhibitor (NCT03213678 and NCT03155620), have shown promising results (Figure 4). Finally, the direct inhibition of GLI through the use of GLI antagonist 61 (GANT61) or arsenic trioxide (ATO) has shown great promise against therapy-resistant medulloblastoma [27].

The ABCB receptor family, which act as ATP-dependent multidrug pumps, are also key regulators of chemoresistance in SHH medulloblastoma. The ABCB1 receptor, primarily responsible for promoting vincristine resistance, is positively associated with upregulation of Y-box binding protein 1 (YB-1). YB-1 is also implicated in enhancing the metastatic potential in various other cancers such as hepatocellular carcinoma, renal cell carcinoma and breast carcinoma [110,111,112]. In vitro analysis of YB-1 suppression via BRD7 interaction demonstrated diminished metastatic potential of breast cancer cell lines [112]. Taylor et al. conducted a cell culture-based analysis of YB-1 knockdown in SHH and Group 3 resistant medulloblastoma cell lines, via shRNA lentiviral transfection (Table 2), and observed a significant decrease in the invasive potential and increased sensitivity to vincristine, Panobinostat (a histone deacetylase inhibitor), and JQ1 (a BET bromodomain inhibitor) [35]. Additionally, BET inhibitors can also successfully suppress SMO signaling via GLI inhibition [113]. Trametinib, a MEK inhibitor, has shown encouraging results in primary studies targeting resistant SHH medulloblastoma [114]. In a parallel fashion, CK2 upregulation, associated with poor prognosis in various malignancies, has been assessed, and CX-4945 was identified as the first orally bioavailable selective CK2 inhibitor to be translated into clinical trials [115,116,117]. Rodent model-based studies targeting alternate tumor growth pathways reveal that transfection of cisplatin-resistant mouse medulloblastoma models with miR-29c-3p downregulates the lncRNA CRNDE expression to increase drug sensitivity, induce apoptosis, and inhibit metastasis [118] (Table 2).

Additionally, inhibiting the previously discussed molecular targets involved in the anti-apoptotic pathways has been explored as a potential strategy to attenuate treatment resistance and promote cell death. Herein, the anti-apoptotic proteins, BCL-XL and MCL-1, have emerged as tractable targets. In specific, the BCL-XL/BCL-2 inhibitor, ABT-263 (Navitoclax), has been shown to induce cell death in medulloblastoma [119]. Additionally, BCL-CL inhibition has shown a synergistic action with the anti-mitotic agent MLN8237 [120] and HDAC inhibitors [121]. In a similar fashion, the MCL-1 inhibitor, A-1210477, showed significant synergism with the GLI-1 inhibitor, GANT61, to induce apoptosis in SHH-driven medulloblastoma [122]. However, the therapeutic effects of these agents, especially ABT, might be hindered by their poor BBB permeability [123]. Hence, further investigation into enhancing their delivery to brain tumors is needed before they can move to clinical trials.

**Table 2 cancers-16-02249-t002:** Summary of the non-coding RNAs that have been shown to decrease treatment resistance in medulloblastoma.

Non-Coding RNA	Target	Effect	Reference
shRNA	YB1	Suppress invasion and increase sensitivity to vincristine, Panobinostat, and JQ1	[35]
miR-29c-3p	-	Augment drug sensitivity, promote apoptosis, and inhibit metastasis	[118]
siRNA	SMO	Induce apoptosis in SHH-activated medulloblastoma cells	[100]
miR-23a-3p	GLS	Reverse metabolic changes and increase sensitivity to cisplatin	[124]

shRNA, short hairpin RNA; siRNA, small interfering RNA; SMO, smoothened; GLS, glutaminase.

### 4.3. Targeting Cancer Stem Cells

Stem cells pose a unique challenge when it comes to solid CNS tumors. Singh et al. reported that these stem cells or tumor initiation cells have remarkable properties of self-renewal, proliferation, and differentiation. They also identified important markers of stemness such as CD133, SOX2, Musashi1, BMI1, etc. [65]. Huang and colleagues reported the signal transducer and activator of transcription 3 (STAT3) as one of the major pathways being upregulated in CD133+ cells [125]. STAT3 inhibitors, cucurbitacin I or celecoxib, can attenuate the stem-like capabilities of CD133^+^ MBSCs and enhance the sensitivity of CD133+ MBSCs xenografts to both chemotherapy and radiation therapy [126,127]. Additionally, sodium butyrate (NaB) is a potent inhibitor of medulloblastoma stem cell expression [128]. In fact, NaB-mediated inhibition of histone deacetylase (HDAC) and ERK/MAPK signaling in two varied medulloblastoma cell lines exhibits downregulation of stemness markers CD133 and MBI1 and reduced cell survival and proliferation [129]. Mithramycin, a SOX2 inhibitor, has been shown to inhibit SHH medulloblastoma tumor growth in culture and is a favorable therapeutic agent [130]. Additionally, JQ1, a BRD4 inhibitor, has also shown promising results in suppressing the expression of stemness markers SOX2, Nestin, and Nanog in medulloblastoma stem cells and inhibits organoid formation in vitro and growth of implanted cells in vivo [71]. Additionally, inhibition of the previously discussed NOTCH pathway in group 4 medulloblastoma has piqued interest from various research groups. Huang et al. demonstrated that NOTCH inhibition through *γ* secretase inhibitors cause inhibition of the HIF1*α* mediated stem cell differentiation and has the potential to improve disease prognosis [125]. Furthermore, Frasson et al. found that the inhibition of the PI3K signaling pathway suppresses the proliferation of CD133+ medulloblastoma stem cells and promotes apoptosis in these cells [131]. Interestingly, these stem cells were also found to be more sensitive to PI3K inhibition compared to the remaining cell population in medulloblastoma tumors [131].

### 4.4. Disrupting Metabolic Patterns

Several metabolic pathways have been investigated as tractable targets to offset metabolic reprogramming and overcome treatment resistance. Ge et al. conducted an in vitro analysis of patient-derived medulloblastoma tumor cell samples and established that the long non-coding RNA nuclear paraspeckle assembly transcript 1 (NEAT1), which is overexpressed in various tumor cells [124,132,133,134], is associated with cisplatin resistance in medulloblastoma. NEAT1 promotes the production of glutaminase (GLS) in tumor cells, which contributes to a shift in the glucose and glutamine metabolic pathways. This Warburg Effect [135] leads to greater survival in a hypoxic environment and drug escape in these cells. They also concluded that inhibition of GLS production through extrinsic administration of miR-23a-3p results in increased sensitivity to cisplatin and better therapeutic outcomes [134]. Other metabolic targets in resistant medulloblastoma include the nucleoside biosynthesis pathway. Recent studies conducted in group 3 MYC-amplified medulloblastoma elucidate that reprogramming of the purine biosynthesis and energy pathways are characteristic findings in resistant medulloblastoma, thus proving to be potential molecular targets [50]. Purine analogues fludarabine, cladribine, clofarabine, and 8-azaguanine act synergistically with the conventional VECC (Vincristine, Etoposide, Cisplatin, and Cyclophosphamide) drug cocktail in non-WNT medulloblastoma to sensitize cells to chemotherapy [136]. Furthermore, targeting energy metabolism pathways in the hypoxic tumor microenvironment has proven to be a promising course of treatment. In this context, phenformin, an inhibitor of the mitochondrial glycerophosphate dehydrogenase, leads to decreased glucose utilization in SHH and group 3 tumor cells in the hypoxic microenvironment, hence reducing survival [137]. Similarly, GNE-140, a lactate dehydrogenase A and B (LDHA/LDHB) inhibitor, induces regression of medulloblastoma group 3 organoids in culture [138]. Finally, as discussed previously, enzymes responsible for the production of SMO-activating lipids confer resistance in medulloblastoma cells and are recent therapeutic targets. Latest investigations have identified HSD11β2 as one such enzyme, and its inhibition with carbenoxolone (CNX) enhances the efficacy of the CDK4/6 inhibitor abemaciclib [92].

### 4.5. Harvesting the Potential of the Immune System

#### 4.5.1. Chimeric Antigen Receptor T-Cells (CAR T-Cells)

The status of the brain as an immune-privileged organ renders medulloblastoma cells unsusceptible to immune surveillance by peripheral T-cells, macrophages, NK cells, etc. On top of that, the presence of tumor-associated macrophages (TAMs) in the niche tumor microenvironment have previously been associated with detrimental effects like promoting growth, invasion, and metastasis through the secretion of pro-inflammatory cytokines including CCL-2 and TGF-β in solid CNS tumors like glioblastoma. Paradoxically, TAMs in medulloblastoma have been associated with restricted tumor growth and overall better prognosis [139,140]. Hence, complementing the anti-tumor actions of TAMs, the delivery of genetically modified CAR T-cells targeting specific tumor markers is a prospective therapeutic approach. Clinical trials targeting the expression of B7-H3 in various CNS tumors and IL13Ralpha2-CAR T, a brain tumor-specific CAR-T cell (NCT04185038, NCT04661384), are currently in phase I stage. HER2 expression is particularly associated with group 3 medulloblastoma, and CSF administration of CAR-T cells targeting has shown to be an effective therapeutic modality [141] (NCT03500991).

#### 4.5.2. Oncolytic Virotherapy

Oncolytic viruses are designed to specifically infect and kill tumor cells. They cause epitope spreading which augments the inflammatory response and immune cell infiltration into the tumor microenvironment. Usually DNA viruses like CMV, polio virus, parvovirus, etc. are used for therapy. A Phase 1b clinical trial is being conducted with the aim of investigating the safety of oncolytic poliovirus therapy administered via convection-enhanced delivery into the brain in a sample of pediatric patients with different types of brain tumors, including medulloblastoma (NCT03043391). Another trial is assessing the safety of an oncolytic herpes simplex virus as an individual therapy or when combined with radiation therapy (single dose of 5 Gy) in patients with recurrent or progressive cerebellar brain tumors (NCT03911388).

#### 4.5.3. Cancer Vaccines

Vaccines for various cancers have been under development for a long time with little success. Cancer vaccines can be DNA, RNA, or peptide based [142,143]. A phase III clinical trial involving patients with newly diagnosed and recurrent GBM recently revealed notable survival advantages for those who underwent autologous tumor lysate-loaded dendritic cell vaccination alongside standard treatment compared to those receiving standard treatment alone (NCT00045968). Interestingly, two phase I studies (NCT03299309 and NCT03615404) are currently evaluating cytomegalovirus-based vaccines, establishing the applicability and safety of this therapeutic modality for recurrent malignant glioma and medulloblastoma in pediatric populations. Another approach of vaccine-induced robust immune response is the use of tumor stem cell antigen-loaded dendritic cells, which is also being studied currently (NCT01171469).

#### 4.5.4. Immune Checkpoint Inhibition

Immune checkpoint inhibitors (ICIs) prevent the progressive enfeebling of the body’s immune response upon continual exposure to tumor antigens. In fact, ICIs are mainly antibodies designed to alleviate T-cell suppression and restore function by obstructing their binding to a specific checkpoint ligand. Inhibitors targeting cytotoxic T-lymphocyte antigen 4 (CTLA-4) and programmed death ligand 1 (PD-L1) are commonly utilized in cancer immunotherapy [143]. Drugs aimed at newly identified antigens, such as B7-H3, CD40/CD40L, IDO1, and mucin domain 3 (TIM-3), are showing promising outcomes in both preclinical and clinical studies for brain tumor treatment [144,145]. Numerous clinical trials investigating checkpoint inhibitor drugs in medulloblastoma patients are currently underway. PD-1/PD-L1 inhibitors, such as nivolumab, pembrolizumab, and durvalumab, are being individually explored as monotherapies in phase I/phase II trials (e.g., NCT03173950, NCT02359565, and NCT02793466). Additionally, 131I-omburtamab, which is a radiolabeled antibody, has made it to clinical trials as a form of radioimmunotherapy (e.g., NCT04743661 and NCT05064306).

#### 4.5.5. Adoptive Natural Killer Cell Therapy

Natural Killer (NK) cells have gained wide attention due to their potential in eliciting anti-tumor effects and aiding in generating a significant immune response. In the context of medulloblastoma, it has been shown that NK cells have the ability to exert cytotoxic effects against both CD133+ and CD133− medulloblastoma cells, which carry ligands that can activate DNAM1, NKG2D, and other receptors on the surface of NK cells [146]. The feasibility and safety of adoptive NK cell therapy has been confirmed in a phase I clinical trial of ex-vivo expanded natural cells that have been injected intraventricularly in pediatric patients with medulloblastoma and ependymoma [147]. Further studies are needed to confirm the efficacy of this therapy and expand on its potential.

## 5. Future Directions

As detailed throughout this review, medulloblastoma cells utilize different mechanisms and processes to evade treatment or adapt to it. Hence, it is crucial to identify these mechanisms, investigate the molecular pathways that mediate them, and try to overcome or suppress them. Otherwise, failure to identify and tackle these resistance mechanisms will lead to treatment failure, relapse of the disease, and the emergence of more aggressive and more resistant colonies within the tumor. Therefore, targeting resistance mechanisms should not be a salvage step that is done after this resistance emerges, but it should rather be a prophylactic step in the management of these tumors that utilizes combination therapy and precision medicine to predict potential resistance patterns and prevent their emergence. This requires dedicating more studies to fully uncover the genetic, epigenetic, molecular, and environmental drivers of these resistance mechanisms and to elucidate effective modalities to curb them.

## 6. Conclusions

The present manuscript reviews the different molecular profiles of medulloblastoma subtypes and summarizes the major mechanisms that have been studied in the literature as drivers of treatment resistance. Some of these mechanisms stem from the fact that the medulloblastoma is a malignant tumor that arises in the “privileged” brain, while others are related to unique properties that are intrinsic to this tumor or to a specific subtype. Consequently, many of the solutions that have been explored to overcome this resistance are shared with other types of solid tumors and have shown great promise in different settings, such as BBB disruption and immune-stimulatory therapies. However, other modalities focus on targeting specific pathways that promote medulloblastoma resistance. These include targeting the molecular drivers implicated in stemness, maintaining an undifferentiated state, metabolic reprogramming, DNA repair, resistance to apoptosis, and other malignant features. The literature summarized in this review shows that substantial research efforts have been channeled towards uncovering the mechanisms responsible for treatment resistance in medulloblastoma and investigating potential modalities to undermine these mechanisms. However, more focus should be directed towards advancing these strategies, translating the safe and effective ones into clinical practice, and promoting the upfront targeting of resistance mechanisms to avoid treatment failure and tumor recurrence.

## Figures and Tables

**Figure 1 cancers-16-02249-f001:**
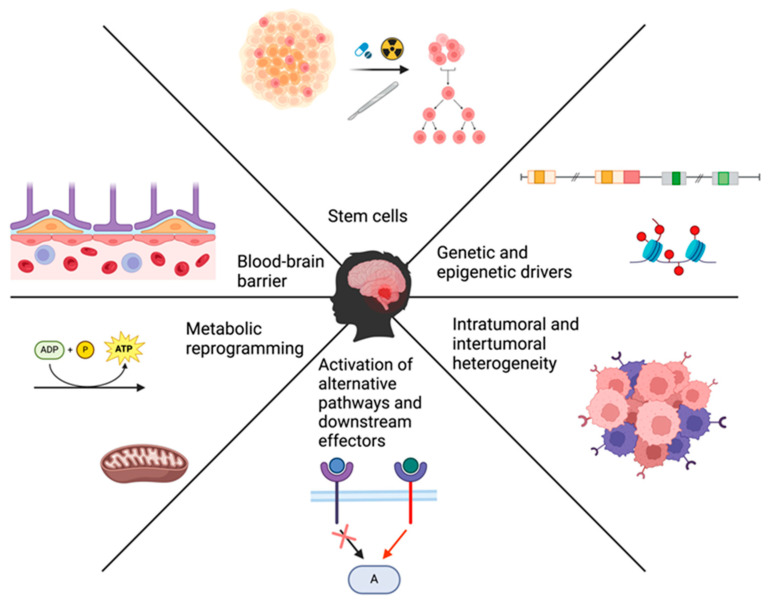
Summary of the various drivers of treatment resistance in medulloblastoma.

**Figure 2 cancers-16-02249-f002:**
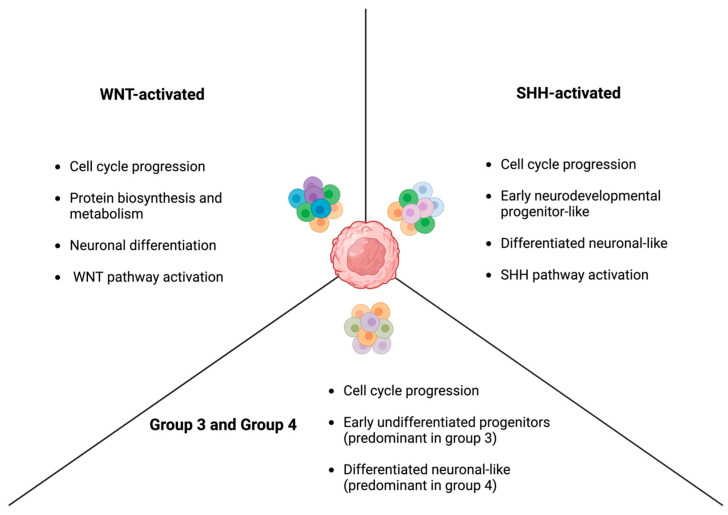
The different cellular states identified through single-cell RNA sequencing within each of the molecular subtypes of medulloblastoma, which shows the considerable intratumoral heterogeneity that can drive treatment resistance.

**Figure 3 cancers-16-02249-f003:**
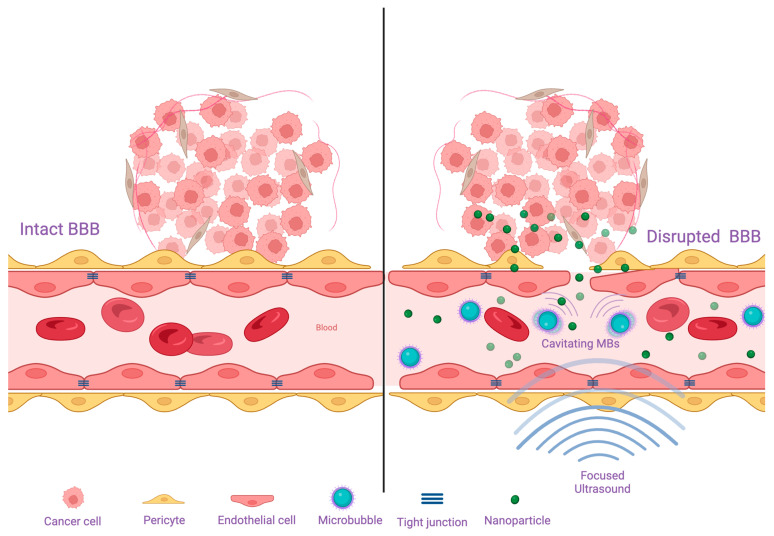
The structure of the BBB and mechanisms to overcome it. (**Left panel**) The BBB is composed of endothelial cells that are connected to each other by tight junctions, and backed up by pericytes and foot processes of astrocytes. This structure impedes the delivery of therapeutic agents to brain tumors. (**Right panel**) Some of the methods that have been used to overcome the BBB are low-intensity focused ultrasound, which can transiently disrupt the integrity of the BBB, and drug-loaded nanoparticles, which have a better ability to penetrate the barrier. The action of focused ultrasound is dependent on the presence of microbubbles that are injected systemically. These microbubbles which should be present in the blood stream respond to the ultrasound waving through cavitation and contribute greatly to the mechanical disruption of the BBB.

**Figure 4 cancers-16-02249-f004:**
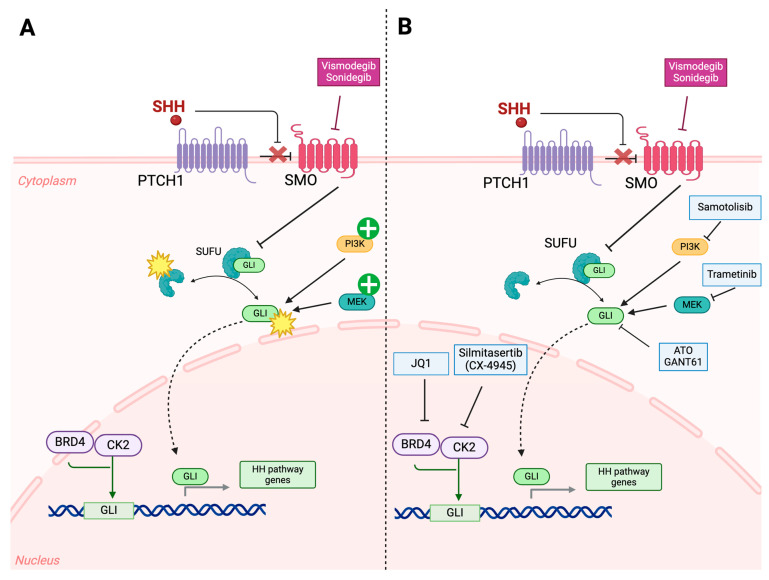
The mechanisms that drive resistance to SMO inhibitors (vismodegib and sonidegib) and potential targeted therapies to overcome them. (**A**) Hedgehog signaling is initiated with the binding of SHH to its receptor PTCH1. This lifts the inhibition that PTCH1 exerts on SMO and allows the latter to initiate a downstream cascade of events. The cascade starts with lifting the control that SUFU exerts over GLI proteins. This SMO-induced activation of GLI promotes the nuclear translocation of this factor and the subsequent transcription of other effectors in the HH pathway that are involved in tumorigenesis. Other mechanisms that are involved in the activity of GLI proteins are the epigenetic regulators that promote their expression, such as BRD4 and CK2. Vismodegib and sonidegib are used to inhibit the activation of the SHH pathway through their action as direct inhibitors of SMO. The mechanisms implicated in the treatment resistance against these SMO inhibitors mainly involve mutations in proteins, such as SUFU and GLI, that are part of the SHH pathway downstream of SMO or the convergence of other overactivated pathways, such as AKT/PI3K and MEK/ERK, on major players of the SHH pathway. (**B**) Targeting the aforementioned mechanisms and molecular drivers can abrogate the treatment resistance to SMO inhibitors when used as a combination therapy. This includes the use of direct GLI inhibitors, such as GANT61 and ATO, or inhibitors of the alternative overactivated pathways, such as the PI3K inhibitor, samotolisib, and the MEK inhibitor, trametinib. Additionally, targeting the epigenetic regulator of GLI expression can also be utilized through the inhibition of BRD4 using JQ1 and CK2 using silmitasertib. ATO, arsenic trioxide; BRD4, bromodomain 4; CK2, casein kinase 2; GANT61, GLI antagonist 61; GLI, glioma-associated oncogene; HH, hedgehog; PTCH1, patched 1; SHH, sonic hedgehog; SMO, smoothened.

**Table 1 cancers-16-02249-t001:** The different epidemiologic characteristics, genetic profiles, and histologic appearance of the four molecular subtypes of medulloblastoma.

	WNT	SHH	Group 3	Group 4
Prevalence	10%	30%	25%	35%
Peak Incidence (age group)	10–12 years	<3 years and >17 years	3–5 years	5–10 years
Prognosis	Good	Good for MBEN	Poor	Intermediate
5-year survival	>90%	70%	50%	75%
Predisposing Genes	*CTTNB1*, *APC*	*SUFU*, *PTCH1*, *PALB2*, *BRCA2*, *TP53*	*PALB2*, BRCA2	
Histologic Presentation	Classic	Classic, D/N, LC/A, MBEN	LC/A	
Metastasis at diagnosis	Rare (5–10%)	Rare (9–30% depending on subgroup)	Very frequent (around 50%)	Frequent (35–40%)

SHH, sonic hedgehog; D/N, desmoplastic/nodular; LC/A, large cell/anaplastic.

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
