# Peer review of "Overcoming Treatment Resistance in Medulloblastoma: Underlying Mechanisms and Potential Strategies"

_cancers, 2024, doi:10.3390/cancers16122249_

Round 1

Reviewer 1 Report

Comments and Suggestions for Authors

In their manuscript, Slika et al. review the most recent mechanisms that have been correlated to the intrinsic or acquired resistance to treatments in medulloblastoma. After a proper initial description of the most relevant molecular features of medulloblastoma subgroups and their expected discrepancies in terms of activated pathways and patient outcomes, authors describe several mechanisms that have been recently reported for their involvement in medulloblastoma aggressive behaviour and treatment resistance. Moreover, they summarize the most relevant therapeutic approaches recently reported to overcome resistance in these tumors, including relatively experimental approaches, together with the ongoing trials in the field.

The manuscript is quite comprehensive; however, I suggest authors to consider these revisions to increase the value of their work, in particular:

1.    I suggest authors to add more detailed information on patient outcome, survival, metastatic status, etc. for group 3 and 4 medulloblastoma tumors. Indeed, there is an evident unbalance in the information provided for WNT and SHH medulloblastoma tumors relative to group 3 and 4.

2.    Table 1 should be integrated with more detailed information. As it is, the table is poorly useful since it does not include patient survival, detailed percentages of metastasis at diagnosis, mean age of patients from different subgroups, etc.

3.    Authors describe the role played by the BBB in sustaining therapy resistance and particularly comment on the leaky nature of WNT medulloblastoma BBB. What about the other subgroups? Based on the study from Genovesi et al, 2021 (REF 27) and data provided by others, I would integrate this paragraph with some additional information on the main BBB alterations occurring in non-WNT tumors.

4.    Authors properly cited the potential role played by anti-apoptotic proteins in sustaining therapy resistance in medulloblastoma. In this context, I suggest them to also include a comment to their potential inhibition as a relevant therapeutic strategy, including the challenge of BBB penetration for these drugs.

5.    In section 4.3 authors suggest some strategies to target cancer stem cells-dependent resistance in medulloblastoma. Moreover, in several paragraphs of the manuscript they highlight the potential role played by PI3K signaling in sustaining therapy resistance. For these reasons, I suggest authors to cite the study from Frasson, C et al (2015) in which a selective eradication of the CD133+ cancer stem cell counterpart has been achieved in medulloblastoma cells through inhibition of PI3K doi: 10.1155/2015/973912.

6.    There has been a mistake in citing the study from Mariotto et al. 2023 (ref 125) since they employed a therapeutic combination of antimetabolites with VECC in “non-WNT” chemotherapy resistant medulloblastoma models and not in WNT medulloblastoma as reported in the manuscript.

Minor comments:

1.    The relevant study from Pham et al. (line 460) has not been cited in exten in the reference section DOI: 10.1158/1078-0432.CCR-15-0713

Author Response

Response to reviewers:

We thank the editor and reviewers for their time and help in improving our manuscript.

Reviewer 1:

In their manuscript, Slika et al. review the most recent mechanisms that have been correlated to the intrinsic or acquired resistance to treatments in medulloblastoma. After a proper initial description of the most relevant molecular features of medulloblastoma subgroups and their expected discrepancies in terms of activated pathways and patient outcomes, authors describe several mechanisms that have been recently reported for their involvement in medulloblastoma aggressive behaviour and treatment resistance. Moreover, they summarize the most relevant therapeutic approaches recently reported to overcome resistance in these tumors, including relatively experimental approaches, together with the ongoing trials in the field.

Thank you for taking the time to read our manuscript and for your valuable feedback.

The manuscript is quite comprehensive; however, I suggest authors to consider these revisions to increase the value of their work, in particular:

  1. I suggest authors to add more detailed information on patient outcome, survival, metastatic status, etc. for group 3 and 4 medulloblastoma tumors. Indeed, there is an evident unbalance in the information provided for WNT and SHH medulloblastoma tumors relative to group 3 and 4.

Thank you for pointing this out. We agree that the sections on group 3 and group 4 required more information on the prognoses and outcomes. We have added information on the 5-year survival, metastasis rates, and genetic landscape of the subgroups (α, β, and γ) in each of these groups (lines 158-163 and 179-183), just like those discussed for the WNT and SHH groups.

  1. Table 1 should be integrated with more detailed information. As it is, the table is poorly useful since it does not include patient survival, detailed percentages of metastasis at diagnosis, mean age of patients from different subgroups, etc.

Thank you for your suggestion. We have updated Table 1 to make it more comprehensive and informative for the readers. We have added information on the 5-year survival rate, percentages of metastasis at diagnosis, and the age groups that are frequently affected by each subtype.

  1. Authors describe the role played by the BBB in sustaining therapy resistance and particularly comment on the leaky nature of WNT medulloblastoma BBB. What about the other subgroups? Based on the study from Genovesi et al, 2021 (REF 27) and data provided by others, I would integrate this paragraph with some additional information on the main BBB alterations occurring in non-WNT tumors.

Thank you for your suggestion. Indeed, important insights have been found regarding the BBB characteristics in the different subtypes of medulloblastoma, which are worth discussing in our manuscript. As per your valued comment, we have added more information regarding the nature of the BBB in non-WNT tumors (lines 253-264).

  1. Authors properly cited the potential role played by anti-apoptotic proteins in sustaining therapy resistance in medulloblastoma. In this context, I suggest them to also include a comment to their potential inhibition as a relevant therapeutic strategy, including the challenge of BBB penetration for these drugs.

Thank you for pointing this out. We agree that it is important to mention the promising effects of anti-apoptotic pathway inhibitors and the potential obstacles that their clinical use might face. We have added information regarding the preclinical studies that have investigated the use of BCL-XL and MCL-1 inhibitors and mentioned their low BBB permeability (lines 704-715), which is a topic that is worth investigating in future studies.

  1. In section 4.3 authors suggest some strategies to target cancer stem cells-dependent resistance in medulloblastoma. Moreover, in several paragraphs of the manuscript they highlight the potential role played by PI3K signaling in sustaining therapy resistance. For these reasons, I suggest authors to cite the study from Frasson, C et al (2015) in which a selective eradication of the CD133+ cancer stem cell counterpart has been achieved in medulloblastoma cells through inhibition of PI3K doi: 10.1155/2015/973912.

Thank you for your suggestion and for bringing our attention to these interesting findings. We have incorporated the suggested information accordingly into the section on targeting cancer stem cells (lines 740-744).

  1. There has been a mistake in citing the study from Mariotto et al. 2023 (ref 125) since they employed a therapeutic combination of antimetabolites with VECC in “non-WNT” chemotherapy resistant medulloblastoma models and not in WNT medulloblastoma as reported in the manuscript.

We apologize for this mistake. We have corrected the information in the manuscript accordingly (line 762).

Minor comments:

  1. The relevant study from Pham et al. (line 460) has not been cited in exten in the reference section DOI: 10.1158/1078-0432.CCR-15-0713

Thank you for your meticulous review and for pointing this out. We apologize for missing this citation. We have now cited this reference as needed (reference 81, lines 488 and 490).

Reviewer 2 Report

Comments and Suggestions for Authors

NK cells mediated immunotherapeutic approachess have to be tackled in the review

Futur recommendations section is needed to be added seperately in the revised version of the manuscript 

Non-coding RNAs involved in the epigenetic modulation could be summarized in table or figure to be more reader friendly and categorized into miRNAs, lncRNAs and circRNAs

Comments on the Quality of English Language

Minor revisions are needed

Author Response

Response to reviewers:

We thank the editor and reviewers for their time and help in improving our manuscript.

Reviewer 2:

NK cells mediated immunotherapeutic approaches have to be tackled in the review.

Thank you for your suggestion. We agree that the use of adoptive Natural Killer cells is an important immunotherapeutic approach that should be tackled. We have added a section on the efficacy of NK cell therapy and the status of clinical trials on the topic (section 4.5.5., lines 829-839).

A future recommendations section is needed to be added separately in the revised version of the manuscript

Thank you for your suggestion. We have added a separate “Future directions” section as recommended (section 5., lines 841-853).

Non-coding RNAs involved in the epigenetic modulation could be summarized in table or figure to be more reader friendly and categorized into miRNAs, lncRNAs and circRNAs

Thank you for your suggestion. We have summarized the non-coding RNAs that have been shown to play a role against treatment resistance in medulloblastoma (Table 2, line 717). We agree that this way they will be more reader friendly and easier to grasp.

Reviewer 3 Report

Comments and Suggestions for Authors

The authors reviewed the different mechanisms that drive relapse and are im-33 plicated in the occurrence of treatment resistance and discusses them in the context of the latest 34 molecular-based classification of medulloblastoma. Generally, the manuscript was well organized and the results are interesting. It is suggested to accept this manuscript after revision.

Q1, please offer more figures to illustrate the discussions

Q2, please use figures to describe the strategies to overcome bbb and highlight the structure of BBB

Q3, please illustrate the combinatory therapy for disease treatment

Q4, please offer information of image-guided drug delivery for disease treatment

Author Response

Response to reviewers:

We thank the editor and reviewers for their time and help in improving our manuscript.

Reviewer 3:

The authors reviewed the different mechanisms that drive relapse and are implicated in the occurrence of treatment resistance and discusses them in the context of the latest molecular-based classification of medulloblastoma. Generally, the manuscript was well organized and the results are interesting. It is suggested to accept this manuscript after revision.

Q1, please offer more figures to illustrate the discussions

Thank you for your suggestions. In response to the reviewer’s comments 1 and 2, we have added a figure that shows the structure of the blood-brain barrier and demonstrates mechanisms that can disrupt/overcome it. Please check the new “Figure 3” in the manuscript (line 590).

Q2, please use figures to describe the strategies to overcome bbb and highlight the structure of BBB

Thank you for your suggestions. In response to the reviewer’s comments 1 and 2, we have added a figure that shows the structure of the blood-brain barrier and demonstrates mechanisms that can disrupt/overcome it. Please check the new “Figure 3” in the manuscript (line 590).

Q3, please illustrate the combinatory therapy for disease treatment

Thank you for pointing this out. The aim behind Figure 4 (previously figure 3) is to illustrate the importance of combination therapy by demonstrating the wide range of therapeutic agents that can be added to SMO inhibitors to augment their effect and decrease treatment resistance. We have modified the figure and edited the caption to reflect that.

Q4, please offer information of image-guided drug delivery for disease treatment

Thank you for your suggestion. Indeed, image-guided drug delivery is an interesting topic that is worth mentioning in the context of drug delivery to the brain. We have added a paragraph regarding this topic and its potential application in brain tumors (601-608).